# N-Use Efficiency and Yield of Cotton (*G. hirsutumn* L.) Are Improved through the Combination of N-Fertilizer Reduction and N-Efficient Cultivar

Jing Niu [1,2], Huiping Gui [1], Asif Iqbal [1] , Hengheng Zhang [1], Qiang Dong [1], Nianchang Pang [1], Sujie Wang [1], Zhun Wang [1], Xiangru Wang [1], Guozheng Yang [2] and Meizhen Song [1,*]

[1] State Key Laboratory of Cotton Biology, Institute of Cotton Research, Chinese Academy of Agricultural Sciences, Anyang 455000, China; niujing890321@163.com (J.N.); huiping.828@163.com (H.G.); asif173aup@gmail.com (A.I.); zhanghengheng1314@163.com (H.Z.); dongqiang@caas.cn (Q.D.); pang2008cotton@163.com (N.P.); wangsujie951230@163.com (S.W.); wangzhun4869@163.com (Z.W.); wxr_z4317@163.com (X.W.)

[2] MOA Key Laboratory of Crop Eco-Physiology and Farming System in the Middle Reaches of Yangtze River, College of Plant Science and Technology, Huazhong Agricultural University, Wuhan 430000, China; ygzh9999@mail.hzau.edu.cn

[*] Correspondence: songmeizhen@caas.cn

**Abstract:** Nitrogen (N) fertilizer plays a vital role in increasing cotton yield, but its excessive application leads to lower yield, lower nitrogen use efficiency (NUE), and environmental pollution. The main objective of this study was to find an effective method to enhance the NUE in cotton production. A two-year field experiment was conducted by using a split plot design with N rates (N0, 0 kg N ha$^{-1}$; N1, 112.5 kg N ha$^{-1}$; N2, 225 kg N ha$^{-1}$, N3, 337.5 kg N ha$^{-1}$), and cotton cultivars (CRI 69; ZZM 1017; ZZM GD89 and XLZ 30) to evaluate both their individual effect and their interactions on cotton yield and NUE. The results showed that the biomass and N accumulation of four cultivars increased with an increase in N rate, whereas the NUE decreased. Cotton yield increased first and then decreased for CRI 69 and ZZM 1017, while kept increasing from N0 to N3 for ZZM GD89 and XLZ 30. Compared with ZZM GD89 and XLZ 30, CRI 69, and ZZM 1017 showed higher yield, resulted from higher biomass, bolls per plant, and boll weight, especially under low N level. In addition, the CRI 69 and ZZM 1017 had stronger N absorption and transformation capabilities, and showed higher NUE than those of ZZM GD89 and XLZ 30 under the same N rate. The results indicated that CRI 69 and ZZM 1017 show advantages over ZZM GD89 and XLZ 30 in yield, and NUE, especially under low N rate.

**Keywords:** cotton; yield; nitrogen rate; NUE

## 1. Introduction

Cotton (*G. hirsutumn* L.) is the most important fiber crop [1], and China is the largest cotton producer as well as consumer globally [2,3]. Cotton growth is influenced by several factors including genotype, environmental conditions, and management practices. Fertilizer is one of the major inputs in cotton production, especially N, which is one of the limiting factors for yield and quality and is required more than other nutrients [4]. Thus, farmers tend to apply large amounts of N fertilizers to improve growth and productivity and ensure high yield [5]. However, excessive N application results in not only excessive cotton vegetative growth, delayed maturity, and reduction of yield and quality, but also increase of N release and environmental pollution [6].

Appropriate increase of nitrogen fertilizer could increase the accumulation of dry matter and content chlorophyll, while an overdose of nitrogen could lead to an imbalance of carbon and nitrogen metabolism, excessive vegetative growth, late maturity [7,8], and decrease the yield and NUE [9–11]. Previous studies found that nitrogen application

within bounds can improve the lint yield and the fiber quality by increasing the dry matter accumulation and photosynthesis [12,13]. Furthermore, other studies indicated that reducing the application of nitrogen without sacrificing cotton yield under reasonable management is feasible and can improve the NUE [5,14]. The same results were found in other crops [15–18]. Besides, stalk recycling to the field is one of the main approaches to avoid the further increase in fertilizers during production [19–21]. However, the excessive application of nitrogen fertilizer in cotton production has not been ameliorated. Therefore, an in-depth analysis of reducing the use of nitrogen fertilizer effects on N use efficiency is still important for cotton production.

In addition to N availability, the genotype is also a key factor determining the growth rates and grain yields of crops. Therefore, it is feasible for the farmer to give up the concept of high-N fertilizer levels and utilize more environmentally friendly genotypes with low-N fertilizer requirements. Since Harvey first reported the differences in nitrogen uptake and utilization among different maize cultivars in 1939 [22], a lot of work on genotype differences of nitrogen efficiency has made great achievements. So far, numerous studies have found that various crops with different genotypes display variance in nitrogen absorption and utilization of nitrogen [23]. The dry weight, photosynthetic efficiency, and yield of nitrogen efficient cultivars were higher than that of nitrogen inefficient cultivars under the same conditions, such as maize [24–27], rice [28–31], wheat [32–34], and other crops [4,35]. In cotton, predecessors have established the screening and evaluation system of different nitrogen efficiency cultivars at the seedling stage [36]. These studies have laid a certain foundation for the efficient utilization of crop nutrient resources and proved to be a feasible way to improve the utilization efficiency of nutrient resources by using nutrient efficient cultivars. Thus, making full use of the plant's own nutrition genetic characteristics for NUE, economize manure prolific breeding, and reasonable application of nitrogen fertilizer are effective way to improve the efficiency of plant nutrients.

Cultivars of CRI 69 and ZZM 1017 had more dry matter and N accumulation in seedling stage than those of ZZM GD89 and XLZ 30 under either N deficiency or sufficiency through hydroponic experiments [37]. We assume that cultivars of CRI 69 and ZZM 1017 may also show their own advantages in the field, making full use of their own advantage can reduce N input, improve N fertilizer utilization. Therefore, a field experiment was conducted on cotton cultivars with varying NUE under various N levels. The aims of this research is to reveal the differences of different cultivars. The purpose of this research is to reveal the differences of various cultivars through the dry matter, N absorption, and distribution as well as yield, and NUE at four N levels. This study is expected to give more insights on the NUE under field conditions with the aim of optimizing N fertilizer utilization and yield.

## 2. Materials and Methods

### 2.1. Plant Materials, Experimental Design, and Field Management

Four cotton cultivars (*G. hirsutumn* L.) were used (CRI 69 and ZZM 1017 were the high NUE cultivars, HNUEC; ZZM GD89 and XLZ 30were the low NUE cultivars, LNUEC) according to our previous work [37].

Field experiments were conducted at the experimental farm of the Institute of Cotton Research, Chinese Academy of Agricultural Sciences, Anyang, Henan province, China (36°06′ N, 114°21′ E) during the cotton growing seasons of 2018 and 2019. The experimental field has been used for cotton planting in the past 5 years, and cotton straw was returned to the soil after harvest. The chemical properties of the field soil were measured for two years (Table 1). The meteorological data are shown in the attached Figure 1. There was no significant difference in monthly average temperature between two years. However, there was a large difference in rainfall between the two years, especially during the boll period. The total monthly rainfall in the growing season was 12.9 mm, 56 mm, and 45 mm in 2018, 2019 and the last 5 years, respectively.

**Table 1.** Soil physical and chemical properties in 2018 and 2019.

| Year | Organic Matter (g kg$^{-1}$) | Total N (g kg$^{-1}$) | Available N (mg kg$^{-1}$) | Available P (mg kg$^{-1}$) | Available K (mg kg$^{-1}$) | pH |
|------|------|------|------|------|------|------|
| 2018 | 15.49 | 0.87 | 74.00 | 16.76 | 175.51 | 8.66 |
| 2019 | 16.07 | 0.96 | 76.00 | 15.62 | 192.20 | 8.61 |

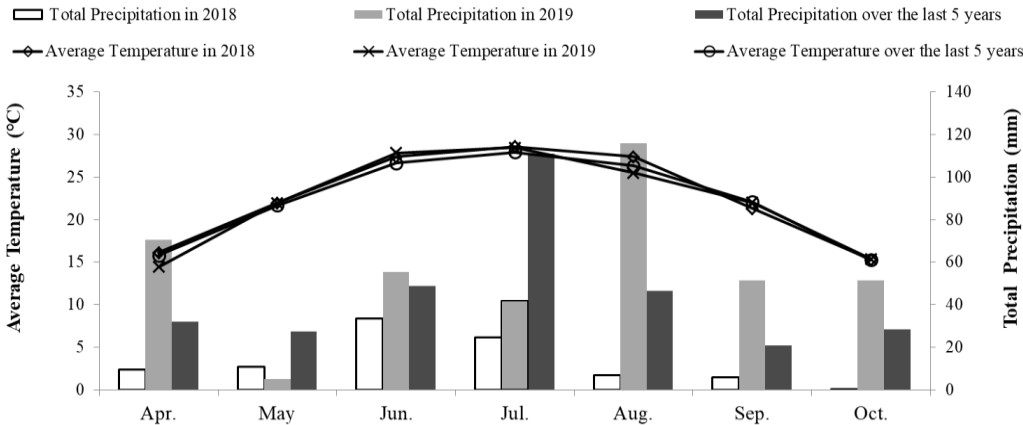

**Figure 1.** Meteorological data during the cotton growth period of the field experiment from May to October on the Anyang of Henan.

The two-year experiment was conducted under a split block experimental design with three replications where the main plot were four N levels (N0, 0 kg N ha$^{-1}$; N1, 112.5 kg N ha$^{-1}$; N2, 225 kg N ha$^{-1}$; N3, 337.5 kg N ha$^{-1}$), and the subplot was cotton cultivar. As for N treatment, half of the target N was used as basal fertilizer and the other half was applied at the full blooming stage by using urea (46.4% N) as N source. In addition, P and K were applied as a basal fertilizer at the rate of 120 kg P ha$^{-1}$ using triple superphosphate (46% $P_2O_5$) and 150 kg K ha$^{-1}$ using potassium sulphate (50% $K_2O$). Experiment plot was 6.4 m wide × 8 m long with 8 rows each spaced 0.8 m apart. A uniform cotton planting density of approximately 52,500 plants per hectare was maintained. The cultivars were planted on 28 April in 2018 and 3 May in 2019, respectively. The cultivation management methods were consistent.

## 2.2. Data Collection

During the growing seasons, plants in the three inner rows of each plot were used for biomass accumulation and N content measurement. At the end of the growing season, the other five rows were used for measuring yield and its components.

### 2.2.1. Yield and Yield Components

Seed cotton of each experimental plot was hand-picked two times (8 and 28 October in 2018, 4 and 31 October in 2019) and weighed after drying to calculate the seed cotton yield. Before harvest, 10 consecutive cotton plants from the inner row were used to determine the yield components. Open bolls were counted for boll number per plant and harvested to calculate the average boll weight. Seed cotton of the 10 plants was ginned to measure lint percentage.

2.2.2. Biomass Accumulation and Partitioning, N Uptake, and N Use Efficiency

The biomass accumulations were measured at 40, 69, 98, and 120 days after emergence (DAE) in 2018 and at 41, 63, 90, and 112 DAE in 2019. Three randomly selected plants from each plot were sampled by uprooting them slowly, and partitioned into vegetative (root, stem (including branches), and leaves), and reproductive organs (buds, flowers, and bolls), and then put in the paper bag. Samples were placed an oven for cell killing at 105 °C for 30 min and then drying at 80 °C to a constant weight before weighing. Biomass partitioning was measured at the boll stage as using the biomass ratio of different plant organs.

With respect to total N uptake, samples of each plant part were milled and screened through a 0.5-mm sieve. The total N concentration was determined using the micro-flow Auto Analyzer 3 (AA3 SEAL, Germany) by the $H_2O_2$-$H_2SO_4$ digestion method and was expressed on a dry weight basis. N accumulation and N utilization-related index were calculated using the following equations [38,39].

$$\text{N accumulation} = \text{N concentration} \times \text{biomass} \tag{1}$$

$$\text{NBE} = (\text{biomass under N treatment} - \text{biomass without N treatment})/\text{N fertilizer rate} \tag{2}$$

$$\text{NAR} = (\text{N accumulation of N treatment} - \text{N accumulation of N0 treatment})/\text{N fertilizer rate} \tag{3}$$

$$\text{NUpE} = \text{N accumulation under N treatment}/\text{N fertilizer rate} \tag{4}$$

$$\text{NUtE} = \text{Yield}/\text{N accumulation of N treatment} \tag{5}$$

$$\text{NUE} = \text{Yield}/\text{N fertilizer rate} \tag{6}$$

$$\text{NHI} = \text{Reproductive organs N content at maturity}/\text{N accumulation of N treatment} \tag{7}$$

where NBE stands for N biological efficiency, NAR is short for N apparent recovery efficiency, NUpE stands for N uptake efficiency; NUtE represents for N utilization efficiency; NUE represents for N use efficiency and NHI is short for N harvest index.

*2.3. Statistical Analysis*

All data were presented as the means of the replicates. Statistical analysis was conducted using SAS 8.1 software (SAS Institute, Cary, NC, USA, 1989). Analysis of variance (ANOVA) was used to test the significance of treatment, group and their interactions using a general linear model. Means were separated using the least significance difference (LSD) tests at the 5% probability level. The maximum yield and nitrogen application amount were calculated through regression analysis of yield and nitrogen fertilizer.

**3. Results**

*3.1. Effects of N and Cotton Cultivar on Biomass Accumulation and Partitioning*

As shown in Table 2, the total dry weight increased as plants grew, following a normal growth curve after emergence. The growth rate of 2019 was faster than that of 2018, especially at the peak bloom stage and boll opening stage, which possibly due to more rainfall 2019 (Figure 1 and Table 2). Averaged across cultivars, the biomass of N1, N2, and N3 treatments were 6%, 15%, and 24% in 2018, and 13%, 30%, and 36% higher in 2019 than N0 treatment at the boll opening stage, respectively.

**Table 2.** Biomass accumulation of cotton cultivars under different N-supply conditions in 2018 and 2019

| Treatment | | Squaring Stage (g plant$^{-1}$) | | Peak Bloom Stage (g plant$^{-1}$) | | Boll Setting Stage (g plant$^{-1}$) | | Boll Opening Stage (g plant$^{-1}$) | |
|---|---|---|---|---|---|---|---|---|---|
| | | 2018 | 2019 | 2018 | 2019 | 2018 | 2019 | 2018 | 2019 |
| N0 | CRI 69 | 12.2bcd | 11.3abc | 38.1bcd | 61.1de | 159.6g | 177.4bcd | 222.0fg | 255.8cde |
| | ZZM 1017 | 11.5cde | 11.7abc | 37.7cde | 61.0de | 153.9g | 183.2bcd | 217.2fg | 258.9cd |
| | ZZM GD89 | 10.8de | 9.3bc | 37.6cde | 50.6fg | 144.2g | 132.0de | 195.0hi | 205.5ef |
| | XLZ 30 | 9.3e | 8.7c | 28.5g | 48.2g | 111.2i | 125.9e | 129.9j | 197.3e |
| N1 | CRI 69 | 13.0ab | 12.2ab | 40.5bcd | 71.1bc | 214.8de | 215.0ab | 283.5bcd | 330.0b |
| | ZZM 1017 | 13.1ab | 11.9abc | 40.0bcd | 70.8bc | 200.1ef | 219.5ab | 269.4e | 329.9b |
| | ZZM GD89 | 11.61bcd | 9.7bc | 38.9bcde | 55.3def | 185.5f | 156.2cde | 239.7f | 253.6cde |
| | XLZ 30 | 9.9e | 9.2bc | 31.9fg | 53.4def | 135.1h | 149.4cde | 174.4i | 236.2def |
| N2 | CRI 69 | 13.5abc | 12.6ab | 45.9ab | 81.5ab | 236.8b | 255.5a | 320.9ab | 389.6a |
| | ZZM 1017 | 13.6ab | 12.4ab | 43.2abc | 82.2ab | 227.6bc | 258.7a | 307.8bc | 391.6a |
| | ZZM GD89 | 11.8bcd | 9.9bc | 40.9bcd | 62.4cde | 205.2de | 185.4bcd | 273.8de | 296.4bc |
| | XLZ 30 | 10.3e | 9.6bc | 33.3e | 60.1cde | 153.4g | 173.7bcd | 208.5hg | 283.1bcd |
| N3 | CRI 69 | 14.1ab | 13.0a | 49.1a | 83.9a | 245.9a | 256.0a | 334.2a | 399.2a |
| | ZZM 1017 | 15.0a | 12.6ab | 45.3abc | 84.5a | 233.7b | 261.1a | 320.9ab | 395.4a |
| | ZZM GD89 | 12.0bc | 10.5b | 45.8ab | 67.7cde | 215.3cd | 200.7bc | 280.4cd | 314.0b |
| | XLZ 30 | 10.9de | 9. 9bc | 35.7de | 64.7cde | 164.0g | 191.6bc | 224.3fg | 298.7bc |
| Nitrogen (N) | | | | | | | | | |
| N0 | | 10.9b | 10.2b | 35.5b | 55.2c | 142.3c | 154.6b | 191.0c | 229.4c |
| N1 | | 11.9a | 10.7a | 37.9b | 62.6b | 183.9b | 185.0b | 241.7b | 287.4b |
| N2 | | 12.3a | 11.2a | 40.8ab | 71.6ab | 205.7a | 218.3a | 277.8a | 340.2a |
| N3 | | 13.0a | 11.5a | 44.0a | 75.2a | 214.7a | 228.3a | 290.0a | 351.8a |
| Cultivar (C) | | | | | | | | | |
| CRI 69 | | 13.2a | 12.3a | 43.4a | 74.4a | 214.3a | 227.0a | 290.2a | 343.6a |
| ZZM 1017 | | 13.3a | 12.2a | 41.5a | 74.6a | 203.8a | 230.6a | 278.8a | 343.9a |
| ZZM GD89 | | 11.6ab | 9.9b | 40.8a | 59.0b | 187.6ab | 168.6b | 247.2b | 267.4b |
| XLZ 30 | | 10.1b | 9.4b | 32.3b | 56.6b | 140.9b | 160.1b | 184.3b | 253.8b |
| Year (Y) | | <0.0001 | | <0.0001 | | 0.002 | | <0.0001 | |
| N | | <0.0001 | | <0.0001 | | <0.0001 | | <0.0001 | |
| C | | <0.0001 | | <0.0001 | | <0.0001 | | <0.0001 | |
| Y*N | | 0.197 | | <0.0001 | | 0.419 | | 0.017 | |
| Y*C | | 0.066 | | <0.0001 | | <0.0001 | | <0.0001 | |
| N*C | | 0.982 | | 0.750 | | 0.328 | | 0.125 | |
| Y*N*C | | 0.944 | | 0.969 | | 0.999 | | 0.950 | |

Note: The means followed by different letters are significantly different at the 0.05 probability level within a column.

N rate and cotton cultivar have significant effects on biomass accumulation of all cotton growing stages (Table 2). Biomass accumulation significantly increased with an increase in N rate from N0 to N2 (0−225 kg N ha$^{-1}$) at the boll setting and opening stage. However, there was no apparent difference between N2 and N3 treatments. In addition, the difference of dry matter accumulation among different N rates increased gradually with the advance of cotton growth period. As refers to cotton cultivars, high NUE cultivars CRI 69 and ZZM 1017 accumulated more biomass than that of low NUE cultivars ZZM GD89 and XLZ 30 at any growth stage. Moreover, the biomass difference between the two cotton types increased as plants grew (Table 2).

*3.2. Effects of N and Cotton Cultivar on Yield and Yield Components*

The yield and yield components varied significantly across years (except boll numbers), cultivars and N fertilizers. More importantly, the N by cultivar interactions effects on yield were significant (Table 3). Under the lower N rate (N0), the plant produced the least bolls, and with the increase of N rate, the boll number increased. The boll number of N0 treatment were only 16.8 and 14.8 per plant in 2018 and 2019, while the boll numbers of N1, N2, and N3 were 0.4, 2.0, and 3.6 higher than N0 in 2018, and were 2.2, 3.7, and 4.6 in 2019. Additionally, the high N efficiency cultivars had more bolls than low N efficiency cultivars.

Averaged across the same cotton type and N rates, the HNUEC and LNUEC produced 20.4, 16.2 bolls per plant in 2018 and 20.2, 15.6 per plant in 2019 respectively (Table 3).

**Table 3.** Yield and yield components of cotton cultivars under different N-supply conditions in 2018 and 2019

| Treatment | | Boll Number per Plant | | Boll Weight (g) | | Lint (%) | | Seed Cotton Yield (kg hm$^{-2}$) | |
|---|---|---|---|---|---|---|---|---|---|
| | | 2018 | 2019 | 2018 | 2019 | 2018 | 2019 | 2018 | 2019 |
| N0 | CRI 69 | 18.6cde | 16.2ef | 5.2bc | 5.5cd | 40.3bcd | 41.7abc | 4489bcd | 4227cd |
| | ZZM 1017 | 18.4def | 16.3ef | 4.9e | 5.7cd | 38.4d | 41.1abc | 4446cd | 4207cd |
| | ZZM GD89 | 15.8ghi | 14.9fg | 4.9e | 5.1gh | 39.2cd | 39.8c | 3854fg | 3435fg |
| | XLZ 30 | 14.3i | 11.8h | 4.5g | 4.6i | 38.2d | 40.8abc | 3049h | 3123g |
| N1 | CRI 69 | 20.3bcd | 18.8cd | 5.3ab | 5.9bc | 41.2ab | 41.8abc | 4675abc | 4599abc |
| | ZZM 1017 | 18.9cde | 19.3bcd | 5.2bc | 6.0ab | 41.9a | 42.1abc | 465abc | 4669ab |
| | ZZM GD89 | 15.2hi | 15.9ef | 5.0de | 5.2fg | 40.9bc | 42.7abc | 3912ef | 3770ef |
| | XLZ 30 | 14.5i | 13.8g | 4.7f | 5.0h | 38.7d | 42.0abc | 3313h | 3357g |
| N2 | CRI 69 | 20.8abc | 20.3bc | 5.4a | 6.1ab | 39.8bcd | 41.6abc | 4784a | 4692ab |
| | ZZM 1017 | 20.8abc | 22.2a | 5.3ab | 6.0ab | 40.1bcd | 42.8ab | 4723ab | 4797a |
| | ZZM GD89 | 16.4fgh | 17.4de | 5.1cd | 5.4efg | 40.0bcd | 41.8abc | 4241de | 4204cd |
| | XLZ 30 | 16.0fgh | 15.2fg | 5.0e | 5.1gh | 39.3cd | 42.5a | 3695g | 3735ef |
| N3 | CRI 69 | 23.2a | 20.7abc | 5.2ab | 6.3a | 39.6cd | 40.7bc | 4761a | 4680ab |
| | ZZM 1017 | 22.3ab | 21.7ab | 5.3ab | 6.2ab | 40.2bcd | 42.5abc | 4748ab | 4766 a |
| | ZZM GD89 | 17.6efg | 18.6cd | 5.0de | 5.5def | 41.1ab | 40.4bc | 4437cd | 4375bc |
| | XLZ 30 | 17.7efg | 17.1de | 5.0de | 5.3efg | 38.9cd | 40.0bc | 3725g | 3960de |
| Nitrogen (N) | | | | | | | | | |
| N0 | | 16.8b | 14.8c | 4.9b | 5.2b | 39.0b | 40.9b | 3959 c | 3748c |
| N1 | | 17.2b | 17.0b | 5.1ab | 5.5b | 40.7a | 42.1a | 4138b | 4099b |
| N2 | | 18.8a | 18.5ab | 5.2a | 5.7a | 39.8ab | 42.5a | 4361a | 4357a |
| N3 | | 20.4a | 19.3a | 5.1a | 5.7a | 40.0ab | 40.9b | 4418a | 4445a |
| Cultivar (C) | | | | | | | | | |
| CRI 69 | | 20.7a | 20.0a | 5.3a | 6.0a | 40.2a | 41.5ab | 4677 a | 4549a |
| ZZM 1017 | | 20.1ab | 20.4a | 5.2a | 6.0a | 40.2a | 42.1a | 4642a | 4610a |
| ZZM GD89 | | 16.7b | 16.7b | 5.0b | 5.3b | 40.3a | 41.2b | 4111b | 3946b |
| XLZ 30 | | 15.6b | 14.5c | 4.8b | 5.0b | 38.8b | 41.6ab | 3445c | 3544c |
| Year (Y) | | 0.243 | | <0.0001 | | <0.0001 | | 0.034 | |
| N | | <0.0001 | | <0.0001 | | 0.002 | | <0.0001 | |
| C | | <0.0001 | | <0.0001 | | 0.035 | | <0.0001 | |
| Y*N | | 0.600 | | 0.002 | | 0.318 | | 0.138 | |
| Y*C | | 0.144 | | <0.0001 | | 0.135 | | 0.102 | |
| N*C | | 0.897 | | 0.120 | | 0.322 | | 0.002 | |
| Y*N*C | | 0.863 | | 0.404 | | 0.613 | | 0.937 | |

Note: The means followed by different letters are significantly different at the 0.05 probability level within a column.

Boll weight varied among different N rates and cotton cultivars. The interaction effect of year by N and cultivar on boll weight were significant (Table 3). As the N rate increased, the boll weight was increased first and stable, which were indicating that appropriate increase of nitrogen fertilizer could increase the boll weight. However, when the nitrogen application exceeded the threshold, the boll weight would not continue to increase. Averaged across cultivars, the boll weight of N1, N2, and N3 treatments were higher 4%, 6%, and 4% in 2018, and 6%, 10%, and 10% in 2019 than N0 treatment, respectively. Additionally, the mean boll weight of CRI 69 and ZZM 1017 was higher 7%, 8%, 6%, and 5% in 2018 and 13%, 14%, 13%, and 12% in 2019 than that of ZZM GD89 and XLZ 30 at N rates (N0–N3), respectively. This result indicated that the cultivars of CRI 69 and ZZM 1017 could get more boll weight especially at the low N level, and the difference between two different NUE cotton types can be reduced by applying more N (Table 3).

A significant interaction was observed between N rates and cotton cultivars for seed cotton yield during both years. Additionally, after comparing and analyzing the yield component, it was found that the yield difference mainly comes from the boll weight

(Table 3). In this study, the yield of two different NUE cotton types responds differently to nitrogen. The yield of ZZM GD89 and XLZ 30 were the highest under the highest N rate (N3), however, the yield of CRI 69 and ZZM 1017 were the highest under the middle N rate (N2). Furthermore, the yield of CRI 69 and ZZM 1017 were significantly higher than that of ZZM GD89 and XLZ 30, and the difference between two cotton types were reduced as the N increase. Averaged across N rate, the yield of HNUEC cotton was 19% and 18% higher than that of LNUEC in 2018 and in 2019. This result indicated that the LNUEC plants need more N fertilizers to gain an equivalent higher yield compare to HNUEC plants. As expected, as shown in Figure 2, the CRI 69 and ZZM 1017 can obtain the highest yield (4694 kg hm$^{-2}$ and 4789 kg hm$^{-2}$) at the N level of 345 kg N hm$^{-2}$ and 27,070 kg N hm$^{-2}$, while the ZZM GD89 and XLZ 30 needs more N (623 kg N hm$^{-2}$ and 4319 kg N hm$^{-2}$, respectively) to achieve their highest yields (4659 kg hm$^{-2}$ and 8886 6 kg hm$^{-2}$).

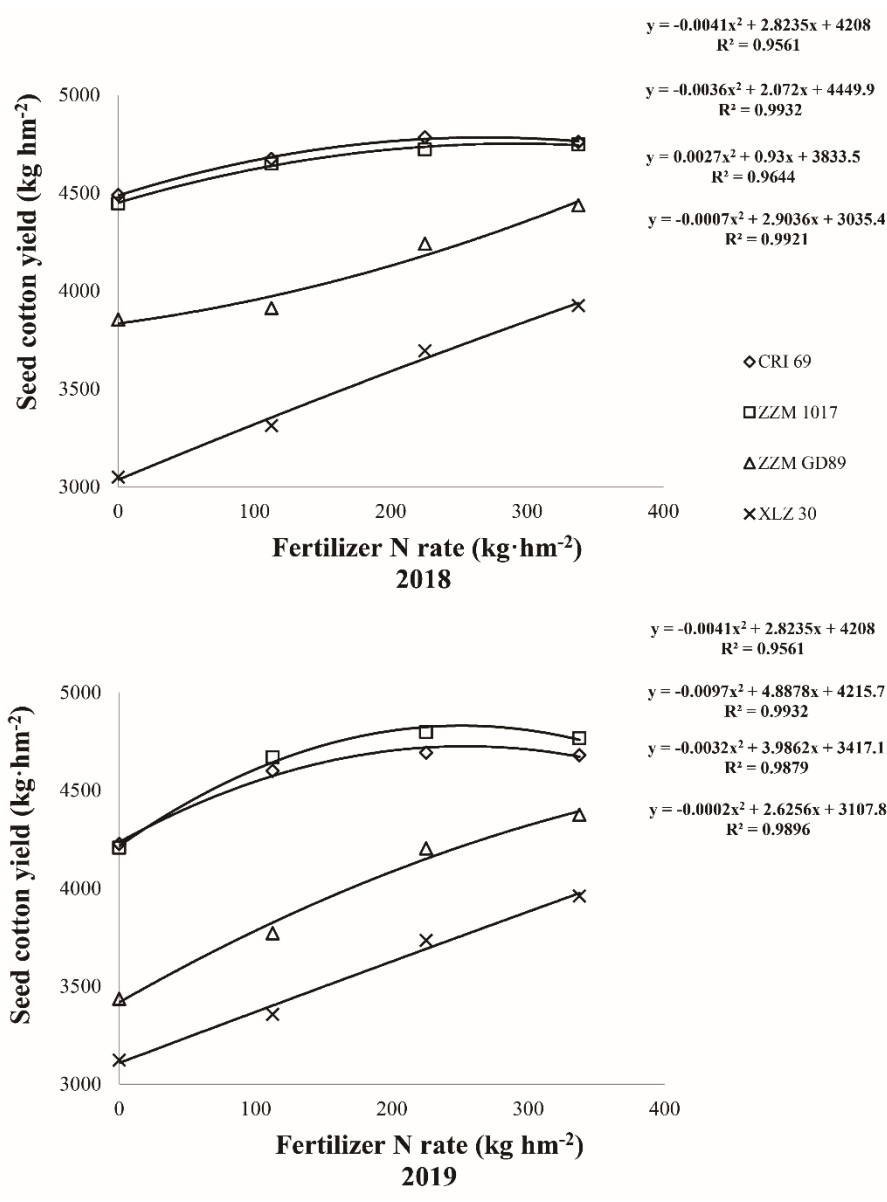

**Figure 2.** Mathematical model between N level and yield.

### 3.3. Effect of N and Cotton Cultivars on N Accumulation and N Utilization Related Traits

A significant difference was observed between two cotton types in some parameters of N accumulation and N utilization. With the increase of N rate, the N content in reproductive organs and total N accumulation of CRI 69 and ZZM 1017 showed a stable trend after

increasing, whereas the relevant traits of ZZM GD89 and XLZ 30 were still increasing (Figure 3). Furthermore, CRI 69 and ZZM 1017 had more N accumulation than that of ZZM GD89 and XLZ 30, the mean N accumulation of CRI 69 and ZZM 1017 was higher 36%, 38%, 41%, and 36% in 2018 and 40%, 48%, 47%, and 39% in 2019 than that of ZZM GD89 and XLZ 30 at N rates (N0–N3), respectively (Figure 3).

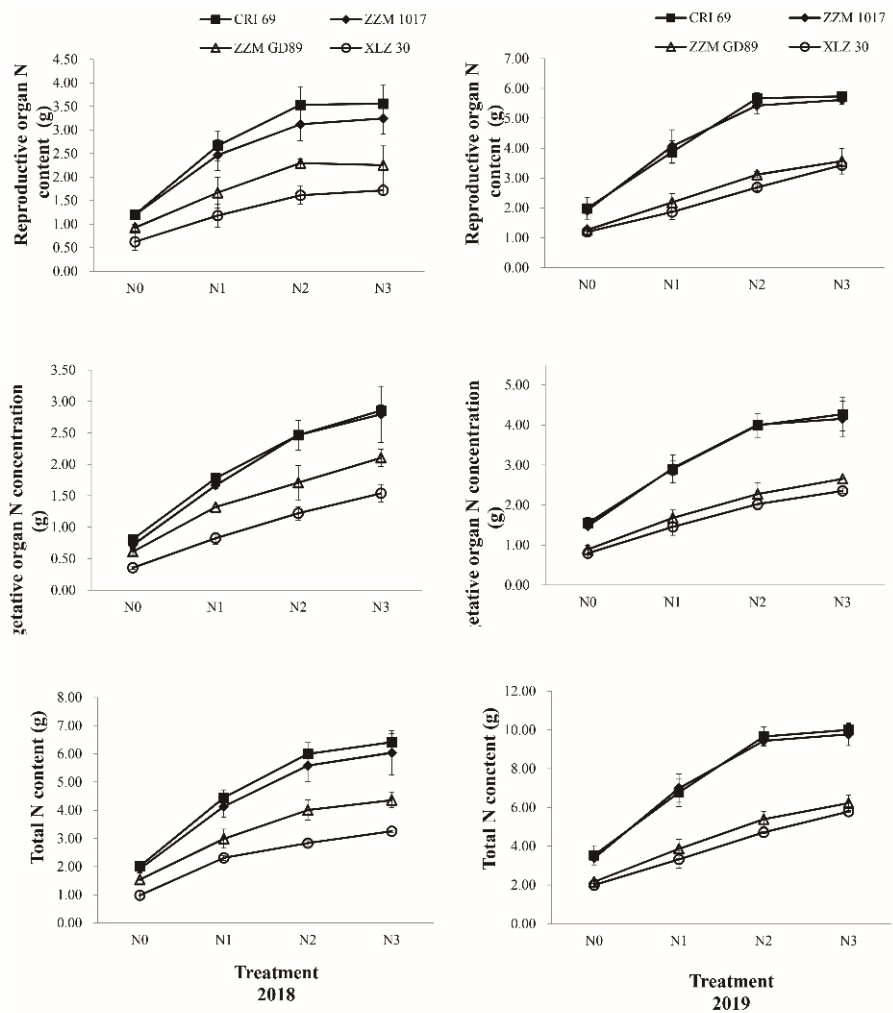

**Figure 3.** Differences in N absorbed of cultivars under different N-supply conditions in the 2018 and 2019.

As shown in Table 4, NBE, NAR, NUpE, NUtE, as well as NUE varied significantly across years, cultivars and N fertilizers (except NHI). Additionally, the interaction effect of cultivar by N on N utilization were significant (except NBE, NUtE, and NHI). All of N utilization decreased as nitrogen increases, (except NHI). In comparison with N1 treatment, the NUE of N2 and N3 was reduced by 48% and 64% in 2018, and 46% and 63% in 2019, respectively. The mean NUE of CRI 69 and ZZM 1017 was higher 21%, 17%, and 14% in 2018 and 18%, 16%, and 11% in 2019 than that of ZZM GD89 and XLZ 30 at N rates (N1–N3), respectively (Figure 3), indicating that the NUE of CRI 69 and ZZM 1017 were significantly higher than that of ZZM GD89 and XLZ 30, and the difference between two different NUE cotton types can be reduced by applying more N.

**Table 4.** N utilization-related traits of cotton cultivars under different N-supply conditions in 2018 and 2019

| Treatment | | NBE (kg kg$^{-1}$) | | NAR (%) | | NUpE (kg kg$^{-1}$) | | NUtE (kg kg$^{-1}$) | | NUE (kg kg$^{-1}$) | | NHI | |
|---|---|---|---|---|---|---|---|---|---|---|---|---|---|
| | | 2018 | 2019 | 2018 | 2019 | 2018 | 2019 | 2018 | 2019 | 2018 | 2019 | 2018 | 2019 |
| N1 | CRI 69 | 28.7a | 34.6a | 113.6a | 151.1b | 2.1a | 3.1a | 20.0de | 12.7c | 41.6a | 39.8a | 0.60a | 0.57ab |
| | ZZM 1017 | 24.3ab | 33.1a | 103.4ab | 168.0a | 1.9b | 3.3a | 21.5cd | 12.4c | 41.3a | 40.2a | 0.60a | 0.58ab |
| | ZZM GD89 | 20.9abc | 23.7b | 67.8d | 79.6d | 1.4c | 1.8c | 25.6ab | 19.2a | 35.7b | 34.5b | 0.56ab | 0.57bc |
| | XLZ 30 | 20.7b | 18.2b | 61.9d | 61.8ef | 1.1d | 1.6d | 27.4a | 20.1a | 29.5c | 30.8c | 0.51ab | 0.56c |
| N2 | CRI 69 | 23.1abc | 31.2ab | 93.0bc | 142.9b | 1.4c | 2.3b | 15.2fg | 9.4d | 21.3d | 21.2d | 0.59a | 0.59a |
| | ZZM 1017 | 21.2abc | 31.0ab | 85.6c | 141.0b | 1.3c | 2.2b | 16.1f | 9.8d | 21.0d | 21.6d | 0.56ab | 0.57ab |
| | ZZM GD89 | 18.4bc | 21.4cd | 57.6de | 75.5de | 0.9e | 1.3e | 20.2b | 15.0b | 18.9e | 18.8e | 0.57ab | 0.58ab |
| | XLZ 30 | 18.3bc | 20.0cd | 43.3f | 63.3ef | 0.7f | 1.1ef | 24.8de | 15.6b | 16.4f | 17.1f | 0.57ab | 0.57ab |
| N3 | CRI 69 | 17.5bc | 22.3cd | 68.5d | 100.6c | 1.0de | 1.6d | 14.1g | 9.0d | 14.1g | 14.1g | 0.56ab | 0.57ab |
| | ZZM 1017 | 16.1bc | 21.2cd | 64.2d | 99.3c | 0.9e | 1.5d | 15.0fg | 9.3d | 14.1g | 14.2g | 0.54abc | 0.58ab |
| | ZZM GD89 | 13.3c | 17.1cd | 43.9ef | 63.2ef | 0.7f | 1.0f | 19.4e | 13.5c | 13.2h | 13.0gh | 0.52cd | 0.57ab |
| | XLZ 30 | 14.7bc | 15.8d | 35.4f | 59.0f | 0.5g | 0.9f | 21.8c | 13.5c | 11.0i | 12.2h | 0.53cd | 0.59a |
| Nitrogen (N) | | | | | | | | | | | | | |
| N1 | | 23.7a | 27.4a | 86.7a | 115.1a | 1.6a | 2.4a | 23.7a | 16.1a | 37.0a | 36.3a | 0.57a | 0.57a |
| N2 | | 20.2b | 25.9a | 69.9b | 105.7a | 1.1b | 1.7b | 19.1b | 12.5b | 19.4b | 19.7b | 0.57a | 0.58a |
| N3 | | 15.4c | 19.1b | 53.0c | 80.5b | 0.8b | 1.2c | 17.9b | 11.3b | 13.2b | 13.3b | 0.53a | 0.58a |
| Cultivar (C) | | | | | | | | | | | | | |
| CRI 69 | | 23.1a | 29.4a | 91.7a | 131.5a | 1.5a | 2.3a | 16.5b | 10.4b | 25.6a | 25.0a | 0.58a | 0.58a |
| ZZM 1017 | | 20.5a | 28.4a | 84.4a | 136.1a | 1.4a | 2.3a | 17.6b | 10.5b | 25.5a | 25.3a | 0.56a | 0.58a |
| ZZM GD89 | | 17.5b | 20.8b | 56.4b | 72.7b | 1.0ab | 1.3b | 21.8a | 15.9a | 22.6b | 22.1b | 0.55a | 0.57a |
| XLZ 30 | | 17.9b | 18.0b | 46.8b | 61.4b | 0.8b | 1.2b | 25.1a | 16.4a | 19.2b | 20.1b | 0.54a | 0.57a |
| Year(Y) | | 0.001 | | <0.0001 | | <0.0001 | | <0.0001 | | 0.581 | | 0.007 | |
| N | | <0.0001 | | <0.0001 | | <0.0001 | | <0.0001 | | <0.0001 | | 0.056 | |
| C | | <0.0001 | | <0.0001 | | <0.0001 | | <0.0001 | | <0.0001 | | 0.057 | |
| Y*N | | 0.768 | | 0.202 | | <0.0001 | | 0.103 | | 0.023 | | 0.017 | |
| Y*C | | <0.0001 | | <0.0001 | | <0.0001 | | 0.003 | | 0.143 | | 0.727 | |
| N*C | | 0.141 | | <0.0001 | | <0.0001 | | <0.0001 | | 0.003 | | 0.143 | |
| Y*N*C | | 0.723 | | <0.0001 | | <0.0001 | | 0.299 | | <0.0001 | | 0.153 | |

Note: NBE, N biological efficiency; NAR, N apparent recovery efficiency; NUpE, N uptake efficiency; NUtE, N utilization efficiency; NUE, N use efficiency; NHI, N harvest index. The means followed by different letters are significantly different at the 0.05 probability level within a column.

## 4. Discussion

### 4.1. Effect of N on Cotton Yield, Yield Components, and NUE

Cotton yield is the result of the coordinated development of yield components such as bolls per ground area, boll weight, and lint percentage. Nitrogen (N) is one of the most important nutrients needed in large amounts for better crop production. Application of optimum N improves various physiological and metabolic processes such as photosynthesis, carbon, and nitrogen metabolism, which is an important limiting factor of high yield and high quality of cotton. Therefore, the application of N fertilizer is one of the important means to increase cotton yield [8]. Many studies have confirmed that a reasonable supply of N nutrition can increase the dry matter and growth rate of cotton at all stages [7]. Additionally, it also improves the dry matter distribution ratio [34,40,41], increases the accumulation and translocation of photosynthetic products [42], and promotes the production [43].

Consistent with previous studies [5,12], increased N rate (N0–N2) significantly improved yield and boll weight. The results indicated that the increased yield was thus attributed to increased boll weight and nitrogen plays an important role in the formation of boll weight and is the main factor affecting yield. However, the biomass and the N

accumulation in plants were highest under the N3 treatment. The studies showed that the increase of nitrogen fertilizer can increase cotton nitrogen absorption, which is beneficial to plant photosynthesis and carbohydrate accumulation [12,34], thereby increasing the biomass, whereas the boll weight was not the highest under the N3. These results proved that excessive use of nitrogen broke the balance between the growth of reproductive and vegetative, leading to a preference for vegetative growth. Besides, we found that the proportion of reproductive organs was the highest under N2 treatment, which was consistent with the research results of Chen et al. [41] and Liu et al. [34] that suitable nitrogen input can promote the transfer of carbohydrates to the reproductive organs and increase economic output. In general, there is positive correlation between yield and biomass, in a certain range. Therefore, yield stabilization can be achieved by maintaining a moderate biological yield and relatively more assimilate distribution on the reproductive organs under the condition of reducing the amount of N fertilizer.

It is well known that the input of N fertilizer increases the N concentration and N accumulation in plants [44–46]. However, NBE, NAR, NUpE, NUtE, and NUE were decreased with an increase in N application [47–49]. We also found that the N content of the reproductive organs increased first and then stable or decreased, but the N content in the vegetative organs N increased, as the N rate increasing, which were proved that the distribution of N assimilate to the reproductive organs reduced under the high N rate or low N rate, it was not conducive to the formation of boll weight and yield. Additionally, the interaction effect of year by N on NUE were significant, proving that the input of nitrogen fertilizer directly affects the nitrogen use efficiency of crops. However, the NUE were much lower than in developed countries such as Europe and America (50–70%), even lower than the average NUE in China (30–35%) [50]. According to China's national conditions, we can only pursue a reasonable NUE with a high yield, instead of blindly pursuing a high NUE to reduce yield. Therefore, it is urgent to find a balance point between nitrogen fertilizer usage, yield, and NUE. In the present study, we found that optimizing the nitrogen allocation ratio of economic organs is a feasible research direction.

*4.2. Effect of Cotton Cultivars on Yield, Yield Components, and NUE*

Many studies on corn and other crops found that the use of N-efficient cultivars can maintains a moderate biological yield and relatively more assimilation distribution on the reproductive organs, to maintain a stable yield, and increase NUE under the conditions of reduced N fertilization. It is may be a way to solve N pollution and is also beneficial to the sustainable development of cotton agriculture. Combining high yield with high NUE is currently challenging. In our environment, poor N use efficiency is found in the cotton.

As discussed earlier that we have identified cotton genotypes with differing NUE and biomass potential [37,51,52]. However, most of these studies focused on the seedling stage plant biomass, photosynthetic activity, and C/N metabolism, and little is known about the genotypic variations in the yield and yield components and NUE. In our study, CRI 69 and ZZM 1017 showed a strong adaptability, especially under low N conditions, so the total N accumulation of CRI 69 and ZZM 1017 are significantly higher than those of ZZM GD89 and XLZ 30, which could promote leaf area development and photosynthetic efficiency [53]. Due to the difference in N uptake and photosynthesis, there was a significant difference in biomass between cultivars. More importantly, the N content of the reproductive organs in CRI 69 and ZZM 1017 were significantly higher than those of ZZM GD89 and XLZ 30 too. The results proved that CRI 69 and ZZM 1017not only have a strong ability to absorb nitrogen, but also have a strong ability to transform into reproductive organs, it could facilitate the DM accumulation and partitioning, resulting in increased yield. The findings agreed with Bange and Milroy [54] and Dai [55]: N-efficient cultivars have a strong adaptability and tend to grow in reproductive organs and were beneficial to the formation of boll number and boll weight, especially under low N conditions. Simultaneously, similar results have been observed in other crops, like, wheat [30], maize [27], corn [34], and poplar [37].

Overall, the yield differences between cultivars maybe mainly caused by nitrogen absorption capacity and nitrogen transferred to reproductive organs. Therefore, we should choose or breed varieties with strong absorption and transfer ability to reduce the N fertilization application under the premise of ensuring yield. Owing to higher N content, and yield, the NUE, NBE, NAR, and NUpE of CRI 69 and ZZM 1017 were significantly higher than those of ZZM GD89 and XLZ 30. These results were consistent with previous studies [36] as well as in with our previous study that ZZM GD89 and XLZ 30 has a poor root system, N uptake and utilization efficiency. The high N uptake and utilization in CRI 69 and ZZM 1017 can be improved photosynthesis and translocation from source to sink tissues which ultimately increase yield, yield components, and NUE [37]. From these results, we concluded that high nitrogen use efficiency varieties have strong nitrogen absorption capacity and transfer ability, especially at low and middle N rates.

Combining high yield with highly NUE is currently challenging. In agricultural production, poor NUE is found in the cotton. Through our research, we found that the N-efficient cultivars can maintain a stable yield and improve the NUE by reducing the N application moderately. At the same time, we need further research to explore the physiological mechanism of efficient and find a highly effective ways to improve the nitrogen utilization efficiency of N-efficient cultivars to improve NUE and cut the nitrogen input. It was in line with the call of the world and China to reduce nitrogen input and N pollution.

*4.3. The Interaction Effect of Cultivar by N on Yield, Yield Components, and NUE*

There were no significant interactions between N and cotton genotypes for biomass, boll number, boll weight, N accumulation, and NUE in both years which agree with previous findings in other crops [18,34]. However, in our study, we have found a significant interaction between N rate and cotton cultivars for yield and NUE. These different conclusions maybe likely associated with crop genetic characteristics, agricultural management and environment. To a certain extent, the results indicated that yield of cotton and the NUE can be improved through the combination of N-fertilizer reduction and N-efficient cultivar.

**5. Conclusions**

In conclusion, the cultivars of CRI 69 and ZZM 1017 (the N-efficient cotton cultivars) have a strong ability to absorb N and transfer N to reproductive organs. Therefore, they produced more the biomass and the boll weight, which were conducive to the formation of output, especially at low N rate. Thus, we should select the N-efficient cultivars to improve the NUE and reduce the N input under the premise of providing guarantee for a yield. More importantly, breeding of high NUtE and NUpE cultivars can reduce both production cost and the environmental concern.

**Author Contributions:** J.N. conducted the main experiment and drafted the manuscript and H.G., H.Z., Q.D., N.P., S.W., and Z.W. helped to collect data. A.I. revised the manuscript and modified the language. X.W. performed part of the statistical analysis and revised the manuscript. M.S., G.Y. and X.W. conceived of the study, participated in its design and coordination, and helped to revise the manuscript. All authors have read and agreed to the published version of the manuscript.

**Funding:** Central Public-Interest Scientific Institution Basal Research Fund (Y2018PT78), Central Public-Interest Scientific Institution Basal Research Fund (1610162020020501), and Agricultural Science and Technology Innovation Program of Chinese Academy of Agricultural Sciences.

**Data Availability Statement:** The data presented in this study are available on request from the corresponding author.

**Acknowledgments:** This work was supported by the Central Public-interest Scientific Institution Basal Research Fund (Y2018PT78), Central Public-interest Scientific Institution Basal Research Fund (1610162020020501), and Agricultural Science and Technology Innovation Program of Chinese Academy of Agricultural Sciences.

**Conflicts of Interest:** The authors declare that they have no known competing financial interests that could have appeared to influence the work reported in this paper.

**Abbreviations**

N, nitrogen; NUE, N use efficiency; NBE, N biological efficiency; NAR, N apparent recovery efficiency; NUpE, N uptake efficiency; NUtE, N utilization efficiency; NHI, N harvest index; DW, dry weight; BN, branch number; BoN, boll number; BW, boll weight; TN, total N content; RN, reproductive organ N content; VN, vegetative organ N content.

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
