# Peer review of "N-Use Efficiency and Yield of Cotton (G. hirsutumn L.) Are Improved through the Combination of N-Fertilizer Reduction and N-Efficient Cultivar"

_agronomy, doi:10.3390/agronomy11010055_

Round 1
Reviewer 1 Report
Your study is adequate in terms of robust data. Your interpretations of the data are questionable and I think that can be improved if you look at the ANOVA table and include the EMS. By doing this you can look at the magnitude of the interactions compared to the main effects. Based on the data in the tables I think many of the interactions are insignificant compared to the main effects. Your tables need much improvements. They are too busy . You report too many digits. Most of the data can be reported to the 0.1 and the yield needs only whole numbers. I have made many suggestions and asked many questions on the paper; please address those concerns. You have presented a lot of data and some don't seem relevant nor do they add to the body of knowledge. Please make major changes to improve this paper.

Reviewer 2 Report
The manuscript by Niu et al. “ N-use efficiency and yield of cotton are improved through the combination of N-fertilizer reduction and N-efficient cultivar” report on the 2-year field trials and analysis of the effect of different N-fertilization rates on nitrogen use efficiency, biomass production and allocation in cotton.
The present manuscript makes a contribution in obtaining new insights on the optimization of fertilizer use in relation to cotton yield production, by analysing multiple yield- and N-metabolism-related traits. This is of particular importance for decreasing the negative impact on the environmental and production costs.
Experimental design of the presented paper is correct; the number of analysed traits exhaustive and statistical analysis appropriate. Only the use of the Duncan’s post-hoc test is questioned, as it is no longer recommended. The authors should consider replacing it with e.g. more versatile Tukey HSD test.
However, the manuscript presents some flaws that should be addressed:
- Result’s presentation - at it stands, it is very difficult to follow decriptions due to excessive use of acronyms and redundancy in describing factor effects, which causes an inefficient highlighting of main results. The suggestion for authors is to describe the huge amount of data on a factor-by-factor basis from ANOVA i.e. to start with significant effects of each factor alone (e.g. genotype, N-dose, year) on all traits involved, and then comment significant interactions between the factors. I believe that the authors should mainly focus on the higher order interactions involving N-factor i.e. GxN and GxNxY, as the N use is the focus of this paper
- Moreover, significant higher degree interactions that deserve detailed description should be presented in dedicated graph/tables with its own post-hoc test. Post-hoc tests reported in present tables seem to take account of genotype as a grouping factor only, so when the authors cite these tables (line 20, pp. 8) it is not possible to evaluate properly the effect of interactions.
- Although that the study is extensive, Conclusions (and Discussion in part) remain rather generic and without much applicable data (see lines 131-134, pp 15). The authors should redefine the conclusion section in order to give more practical aspect to their findings and to recommend optimal N fertilization rate for optimal yield.
- The “mathematical model” presented in Fig.2 and Fig. 4 seems regression analysis of two traits plotted together – please adapt the text and comments accordingly throughout the manuscript, and add the description of this analysis in the M&M section
Other minor comments:
- When commenting an interaction, please make sure that you always clearly refer to the corresponding table for better clarity and sustaining of your claims.
- Given the extensive amount of data presented, the authors should comment only the significant results and not the tendencies.
- When not significant post-hoc Duncan’s results are obtained, avoid putting the same letter next to the values/genotype, and leave blank space in table cells – this will reduce the complexity of your table (e.g. see comment in Table 6, but it is valid for all other similar cases).
- Provide also tables with 2-year means of the recorded traits, which can be placed in Supplementary material; this is informative for the overall evaluation of genotype performance
- Figures with function curves (e.g. Fig. 2) lack significance for the R2, please provide.
- There are too many tables and figures in the main text, please consider pacing some of them in the supplementary file
- Check line numbering
See the attached file for other details and examples of suggestions.

Reviewer 3 Report
The manuscript needs to be revamped, from the abstract to conclusion. It seems like the authors did not review their abstract before submitting their work.

Reviewer 4 Report
I have attached some suggested edits on your paper.
My main concern is with the writing style in the results and discussion section. I think this could be improved by moving or deleting some words, to try and be more clear and precise. Some sentences are very difficult to follow. The information is there it just needs a good scientific edit. This would not take very long for an expereinced scientific author, but could take quite a while by someone with less expereince in english scientific writing. I actually gave up reading bits of the discussion as it needs this sort of edit first.
I think you have done the hard part, it just needs some polish and editing.

Round 2
Reviewer 1 Report
I will not put any more work into this review until the authors make major changes. First there needs to be a Combined ANOVA that shows Mean Squares so that the magnitude of interactions can be compared to main effects. I mentioned this in my first review and it was not done. Next the authors do not need to show results by year when there is no interaction with year. Table 3 is not needed since there were no significant main effects. Figure 2 is not needed either since there was not a significant year interaction. The authors make contradictions when the ANOVA says no interactions and then they make a comment about the tendency or... for an interaction. You cannot do this. The rainfall is the total not the average for a month like shown in Figure 1.
I want to see a clean revised manuscript not one with all kinds of strike marks, etc.
You have important data now you need to make it readable and understandable.
Author Response
Comments and Suggestions for Authors
I will not put any more work into this review until the authors make major changes. First there needs to be a Combined ANOVA that shows Mean Squares so that the magnitude of interactions can be compared to main effects. I mentioned this in my first review and it was not done. Next the authors do not need to show results by year when there is no interaction with year. Table 3 is not needed since there were no significant main effects. Figure 2 is not needed either since there was not a significant year interaction. The authors make contradictions when the ANOVA says no interactions and then they make a comment about the tendency or... for an interaction. You cannot do this. The rainfall is the total not the average for a month like shown in Figure 1.
I want to see a clean revised manuscript not one with all kinds of strike marks, etc.
You have important data now you need to make it readable and understandable.
Reviewer comments:
- First there needs to be a Combined ANOVA that shows Mean Squares so that the magnitude of interactions can be compared to main effects. I mentioned this in my first review and it was not done.
Response: Sorry, we forgot to modify in the manuscript last time. I have modified it this time.
- Next the authors do not need to show results by year when there is no interaction with year.
Response: Most indicators have interaction with year (p<0.005). So the data are displayed according to the year.
- Table 3 is not needed since there were no significant main effects.
Response: The reproductive organs are the harvest organs, so we compared the distribution of organs to find out the differences between different types of cotton. Although there was no significant difference, reproductive organ allocation ratio of CRI 69 and ZZM 1017 was higher than that of ZZM GD89 and XLZ 30. But, I accepted your suggestion that Table 3 has been deleted.
- Figure 2 is not needed either since there was not a significant year interaction
Response: There was a significant year interaction shown in Table 4(P=0.034). Figure 2 is reasonable, and the simulation curve of two-year are separated.
- The authors make contradictions when the ANOVA says no interactions and then they make a comment about the tendency or... for an interaction. You cannot do this.
Response: I have modified the results part about the description of no significant difference.
- The rainfall is the total not the average for a month like shown in Figure 1.
Response: The rainfall is the total for a month. I have modified it.
Reviewer 2 Report
It seems that the authors did not employ enough time to think more critically about the results, which remains insufficiently well described. Scientific edit should be significantly improved in order to describe properly the results and highlight the main findings, before re-submitting the manuscript. For example, significance of specific interactions was often generalized for all conditions. The authors are encouraged to check other publications with similar analysis (across different species) and to restyle and rewrite the manuscript to avoid simplistic way of presentation. Moreover, statistical analysis is not properly described and to me it is not clear whether they analysed the two years separately or together. The authors are encouraged to review the statistics and restyle the way of presentation and discussion (some specific suggestions were given in my previous review). The effort to improve the manuscript is evident but not sufficient.
Author Response
Comments and Suggestions for Authors
It seems that the authors did not employ enough time to think more critically about the results, which remains insufficiently well described. Scientific edit should be significantly improved in order to describe properly the results and highlight the main findings, before re-submitting the manuscript. For example, significance of specific interactions was often generalized for all conditions. The authors are encouraged to check other publications with similar analysis (across different species) and to restyle and rewrite the manuscript to avoid simplistic way of presentation. Moreover, statistical analysis is not properly described and to me it is not clear whether they analysed the two years separately or together. The authors are encouraged to review the statistics and restyle the way of presentation and discussion (some specific suggestions were given in my previous review). The effort to improve the manuscript is evident but not sufficient.
Reviewer comments:
- It seems that the authors did not employ enough time to think more critically about the results, which remains insufficiently well described. Scientific edit should be significantly improved in order to describe properly the results and highlight the main findings, before re-submitting the manuscript. The authors are encouraged to check other publications with similar analysis (across different species) and to restyle and rewrite the manuscript to avoid simplistic way of presentation.
Response: Based on the literature review and your suggestions. We analyzed the data from the Nitrogen (N), Cultivar(C), Year(Y) and their interaction. We got the results that yield of cotton and the NUE can be improved through the combination of N-fertilizer reduction and N-efficient cultivar. N-efficient cultivar have a strong ability to absorb N and transfer N to reproductive organs, which produced more the biomass, the boll weight, and were conducive to the formation of output, especially at low and middle N rate.
- Moreover, statistical analysis is not properly described and to me it is not clear whether they analysed the two years separately or together.
Response: In the results section, we analysed the two years separately to the difference between two cultivars and N rate. We analysed the two years together to the trend and the effect of Nitrogen (N), Cultivar(C), Year(Y) and their interaction.
- The authors are encouraged to review the statistics and restyle the way of presentation and discussion (some specific suggestions were given in my previous review). The effort to improve the manuscript is evident but not sufficient.
Response: I have modified the manuscript and re-analyzed the data to you suggestions that I think are feasible and acceptable.
Reviewer 3 Report
The authors need to incorporate all changes suggested in the first review and provide a clean manuscript free of marks.
Author Response
Comments and Suggestions for Authors
The authors need to incorporate all changes suggested in the first review and provide a clean manuscript free of marks.
Response: Thank you. I have revised it based on your suggestion, and provide a clean manuscript.